# A socioscientific issues approach to ninth-graders' understanding of COVID-19 on health, wealth, and educational attainments

**Wardell Powell** [ORCID] *

Department of Education & Social and Behavioral Sciences, Framingham State University, Framingham, MA, United States of America

* wpowell1@framingham.edu

**Data Availability Statement:** All data are within the paper and its Supporting Information files.

**Funding:** The author received no specific funding for this work.

## Abstract

This study investigated the implementation of a curricular unit of instruction designed to immerse rising ninth-grade students socioscientific issues via data collection and analysis of the relationships between health, wealth, educational attainment, and the impact of the COVID-19 Pandemic on their communities. Twenty-six (n = 26) rising ninth-grade students (14–15 years old; 16 girls, 10 boys) participated in an early college high school program operated by the College Planning Center at a state university in the northeastern United States. The findings of this study demonstrate how ninth-graders enhanced their understanding of the relationships between COVID-19 and community health, wealth, and educational attainments. The students also identified from their research data that communities in Massachusetts that are more educated and with more financial resources were less impacted less by the virus.

## Introduction

The COVID-19 Pandemic has brought to light the effects of long-term discriminatory policies and practices that marginalized communities in the United States have endured for decades. While this may come as a surprise to many who live and work outside of these impoverished communities, many public health scholars, activists, and the minoritized populations from these communities are aware from their lived experiences how policies that often create discriminatory practices in health, wealth, and educational attainment generate more harm than good. The disparities in COVID-19 morbidity and mortality remind us once more of the harmful effects of such long-term discriminatory policies on the minority population. In this study, ninth grade students explored how the least educated and most impoverished communities throughout their state have experienced the brunt of the COVID-19 Pandemic. The purpose of this study was to use socioscientific issues as a vehicle to immerse rising ninth-grade students in data collection and analysis of the relationships between health, wealth, educational attainment, and the impact of the COVID-19 Pandemic on their communities. Below is the description of the socioscientific issues curricular unit on COVID-19, the results obtained, and the advocacy in which the students engaged as they sought assistance and policy changes to their communities' economic, educational, and social infrastructure.

**Competing interests:** The authors have declared that no competing interests exist.

## Literature review

There is a close association between race, educational attainment, wealth, and health outcomes in the United States. Studies have shown that minoritized populations in the United States experience higher rates of chronic and acute illnesses than their white counterparts [1]. Similarly, minoritized populations die earlier and have a higher death rate from environmental toxins and a lack of protective health resources, such as education, healthy food choices, clean water, and safe working conditions [1–3]. While many Americans may believe that minoritized populations put themselves in these predicaments, the facts are that minoritized communities tend to have lower performing schools, more liquor stores, more fast food joints, and higher levels of environmental toxins from industries and highways. These problems set the stage for minoritized populations to become more susceptible to diseases and death. In their recent article titled, "Post-Pandemic Science and Education," Blandford and Thorne [4] provided some suggestions on post-pandemic science and science education:

> Science educators, we must institute a nationwide, crash program (as in the post Sputnik era) to infuse society with science-educated citizens who are imbued with curiosity about the world around themselves and understand the nature and power of critical thinking; citizens who have learned to probe in search of truth and be skeptical of assertions made without basis in fact; citizens who have discovered how easy it is to be proved wrong; citizens who appreciate that often the wisest among us are those who have made the most mistakes most rapidly in search of truth or at least of deeper understanding; and citizens who thereby have the tools to distinguish reliable information from propaganda and wishful thinking [4].

To get students to think about how science is impacting their daily lives, science teachers must provide more opportunities in classroom settings where students learn to determine, collect, and analyze data on scientific phenomena that are impacting their communities. This will create more relevance between science and community, and ultimately impact students' critical thinking and decision-making skills [5]. We want citizens to question others when assertations are made without facts. Our students must learn how to identify fallacies that are masquerading as facts. These skills will allow students to become better advocates for their communities and the environment.

The fallout from the COVID-19 Pandemic has shown that a large segment of the American public has turned against science and rational thinking. News reports of school board meetings across the United States have shown that many Americans have embraced propaganda and wishful thinking instead of what science says. They argue that students should not wear masks to protect themselves and others from a respiratory virus, nor should eligible students be vaccinated. Many school board meetings have even turned violent to the point where police officers are needed to escort to safety scientists, school board members, and parents who support masks and vaccination in schools. The high rates of morbidity and mortality of the COVID-19 Pandemic are there for everyone to see; however, some even deny that there is a pandemic.

In a recent NPR report on the Coronavirus Crisis in Louisiana, a well known United States Senator, who is also a medical doctor was asked the following question:

> *Can you tell me what is being done to help the black community in your state right now, which clearly is being disproportionately hit by this disease?*

The Senator provided a troubling response to the above question. He said,

> . . . if you're going to look at the fundamental reason, African Americans are 60% more likely to have diabetes. Now, if you look at the NIH website, that would say that's for obesity, for genetic reasons, perhaps other things [6].

The senator's response that African Americans are more likely to be diabetics is correct. However, he did not outline underlying reasons for the lack of protective health resources in communities of color, such as education, environmental toxins, and a lack of healthy food choices, clean water, and safe working conditions. Instead, the senator subtly stated that the African American community in Louisiana is disproportionately affected by the Coronavirus because of genetic reasons and perhaps other things.

The senator's reasoning as to why African Americans are disproportionally affected by COVID-19 indicates that school science must provide students with the skills to question assertations without facts. The senator was quick to blame those affected the most by the virus rather than giving a reasoned response on what can be done to reduce health disparities in marginalized communities nationwide. The senator's views highlight his belief that social, economic, health, and educational opportunities are the same for all Americans regardless of race, ethnicity, gender, and social class. This deficit perspective has been reported in science education research.

A recent study in the United States that investigated the racial and ethnic disparities in excess deaths during the COVID-19 Pandemic between March and December 2020 reported that 477,200 more deaths occurred than during the same period in 2019 [7]. The study finds that 74% of these deaths were attributed to COVID-19. The study further reports that deaths of African Americans, American Indian/Alaska Native (AI/AN), and Latino men and women were more than double those among White and Asian men and women. The study also reports that non–COVID-19 excess deaths per 100,000 persons were 2 to 4 times higher in African American, AI/AN, and Latino men and women including deaths due to diabetes, heart disease, cerebrovascular disease, and Alzheimer's disease when compared to whites. The study concludes that excess deaths in 2020 resulted in a substantial widening of racial/ethnic disparities in all-cause mortality from 2019 to 2020 [7].

Increased morbidity and mortality among the minoritized population in the United States are not about genetics. Instead, they are a symptom of the racial and ethnic disparities in the United States related to social, economic, health, and educational opportunities. In addition, it is well understood that long-term exposure to environmental hazards, the intensity of hazards, poverty, and governmental neglect result in ill health in populations.

These conditions will result in vulnerability to diseases. Such exposure will contribute to increased morbidity and mortality. The COVID-19 Pandemic reminds us that those who are more vulnerable will be more susceptible. The Senator, who is a medical doctor, and should know better, would suggest otherwise. It is precisely these reasons that school science must be taught for social justice. Science teachers must become comfortable using real-world issues to engage students in learning science for social justice. This method of teaching science is not new. It has been done with students even at the elementary level. For example, Calabrese-Barton & Upadhyay [8] integrated literacy and technology into an instructional unit on the weather for a group of fifth graders. The students gathered and reasoned about real-world evidence as they built a digital story for how younger students might stay safe from weather events while on the playground. Calabrese-Barton and Upadhyay reported that all of the students took a personal stand on how storms form and why it matters when they shared their digital stories. The authors noted that from their music and sound selection to their word and image choices, science—as content, a discourse, and a way of knowing—became a tool they could use to engage others about their world [8].

The disparity in the COVID-19 infection rates among Massachusetts cities and towns and many other places in the United States presents an opportunity for science teachers to use real-world applications in their science classrooms. This type of science teaching and learning will engage students in scientific practices and discourse to develop scientific knowledge that they can use to empower themselves and others around them. In addition, teaching science in this manner equips students with the skills to navigate effectively in a democratic society as they challenge the socio-political context of COVID-19 and other environmental health-related problems that plague marginalized communities in the United States. Finally, science teachers must recognize their obligations to create learning opportunities for their students to draw upon their scientific knowledge as they assess fallacies promoted by others and make reasoned decisions about the impact of inequity on their daily lives. The COVID-19 Pandemic clearly showed how racial and ethnic disparities are associated with social, economic, health, and educational opportunities.

In order for teachers to get students to think about and make reasoned claims and decisions about how racial and ethnic disparities affect their daily lives, science classrooms must become comfortable spaces where social justice becomes part and parcel of science teaching and learning. This will allow students to recognize that institutions in society operate in a manner that promotes and reinforces different forms of inequities [9]. The science classroom should be the place where students are academically empowered with the scientific knowledge that allows them to grow in their abilities to ask difficult questions, which include but are not limited to: Why are there disparities in water and air qualities, and why are there disparities in diseases and health outcomes among people from marginalized backgrounds?

Studies have reported that students from minoritized groups are not attracted to science because they find that the topics covered in their science classroom are not exciting and that these topics are unrelated to their lives [10]. Getting students more enthusiastic about science may require science teachers to provide more learning opportunities that allow students to grapple with real-world scientific issues. Effectively doing so will enable students to develop scientific knowledge that empowers them to be more active in preserving the environment in which they live, as well as opportunities for students to be skeptical when pseudo-science masquerades as real science. Not providing opportunities for students to learn about the impact of scientific issues on their daily lives runs a significant risk of students becoming gullible and easily manipulated into thinking that there are no set scientific practices in science.

The COVID-19 Pandemic is a case in point. The spread of mis-and disinformation related to the Coronavirus contributed to many lives lost due to distrust for scientists, people having negative attitudes towards science and scientific recommendations, distrust for vaccines, and so on. All science teachers, both in and out of school settings, must use the mistrust for science that they witnessed during the COVID-19 Pandemic to implement learning opportunities for students to learn about real-world scientific issues, with the hope of helping students to become competent in constructing, testing, refining, and justifying evidence-based scientific explanations as they make authentic connections between environmental issues and their daily lives [11].

Health threats resulting from social and economic hazards require education in general and school science, in particular, to equip students with the skills to seek solutions to these problems. It is paramount that school science require students to ask questions and define problems, plan, carry out investigations, analyze and interpret data, argue from evidence, and evaluate and communicate findings [12]. These science and engineering practices will help students engage in evidence-based reasoning, and demand others do the same. To do this, students must be exposed to real-world problems in an authentic manner. The use of socioscientific issues in the school curriculum is one method that can help students engage in investigation and discussion of real-world problems that create risks to human health.

## What are socioscientific issues

Socioscientific issues represent real world problems that are undergirded by moral and ethical implications. As defined by Zeilder (2021) and others:

> Socioscientific issues is, de facto, connected to the quality of personal choices about community and global issues. This is what makes socioscientific issues viewed as "Science-in-Context." It requires a blend of socioscientific reasoning, evoking dimensions of understanding and recognizing elements of complexity, inquiry, perspective taking, skepticism and affordances & limitations of science, as well as the exercise of socioscientific perspective taking [13].

Zeidler's definition of socioscientific issues indicates that using this pedagogy creates opportunities for teachers to use this pedagogy to allow students to investigate real-world issues such as the potential relationships between the COVID-19 infection rates, health, wealth, and educational attainment among populations. Currently, confirmed cases of COVID-19 are measured by rates of positive findings among people who have been tested for the virus. Most of the people who are tested for the virus generally get tested because they show symptoms of being infected. Early in the pandemic, many businesses across the United States also mandated testing for employees, both vaccinated and unvaccinated. Due the varied nature of testing, and the reasons for testing, this has contributed to missing data. Therefore, some would argue to the rates of infection might be higher, while others might argue that the rates are lower. Whatever the argument, the students in this investigation used the infection rates that were reported by their state to conduct their arguments on what must be done to curtail the spread of the virus.

## Method

This study employed the use of a critical ethnography. This methodology allows for participatory critique, transformation, empowerment, and social justice [14,15]. In order for students to critique the disparities in health, wealth, and educational attainment, and the relationships to the COVID-19 infection rates in their communities, critical ethnographic methodology was selected because it challenges the status quo and the dominant powers in society [16]. This methodology provides the opportunity for the understanding of oppression, racism, sexism, and classism experienced by minoritized populations [16].

This study was conducted in the summer of 2020. Before any research procedures, my university Institutional Review Board (IRB) reviewed and approved the study. The director of our College Planning Center, and the parents and guardians then gave written consent. The students also signed an assent form to participate in the study.

Note that care was taken to remove all identifying information from the data to protect the participants' privacy in this study.

To better understand the widespread disparities in health, wealth, and educational attainments in communities across Massachusetts, the participants were asked to analyze COVID-19 infections rates in various cities and towns. The participants conducted investigations on assigned cities and towns across Massachusetts to learn about per-capita income, the percentages of the population with and without a high school diploma, and the percentages of the population with a college degree. The participants then analyzed these data to determine patterns and trends. The data generation description below detailed the activities involved in this study. The research questions below and corresponding rationales were used to guide this investigation.

### Research Question 1

1. What do middle school students think about when asked to explain the relationships between health in Massachusetts communities and the COVID-19 infection rates?

**Rationale 1.** It has been reported that the health sector itself has little to no direct control over most of the underlying conditions required for health [17]. Based on the strong association between poverty and health, it is now time for the health sector to implement and enforce standards for health determinants in communities across the United States. This includes, but is not limited to, environmental conditions, food and drug safety, alcohol and tobacco control, and access to health-related education [17]. The COVID-19 Pandemic has shown the results of the absence of a strong association between the health sector and the underlying conditions required for health. We have read news reports throughout this pandemic that the Coronavirus has caused more hardship in marginalized communities and among the minoritized population. One reason for this widespread devastation is the disparities in health among the minoritized population. What students think about the relationships will be highlighted. This study will show the relationships between health and COVID-19 infection rates in Massachusetts cities and towns.

### Research Question 2

2. How proficient are middle school students in using data on COVID-19 infection rates to make evidence-based claims of its relationship to the wealth-health gap among communities in Massachusetts?

**Rationale 2.** Several studies have confirmed links between income and health [18–22]. More and more studies have revealed that greater levels of wealth are a predictor of better health outcomes [23]. In their report titled, "Wealth Matters for Health Equity," Braverman and colleagues stated that the United States now has the greatest economic inequality of any affluent nation, and, despite being among the wealthiest nations overall, ranks at or near the bottom among affluent nations on almost all measures of health [23]. This study provided an opportunity for students to investigate the COVID-19 infection rates and the wealth-health gap among communities in Massachusetts.

### Research Question 3

3. Did the students' investigations of COVID-19 infection rates in Massachusetts cities and towns impact their understanding of the need for using data to inform decision-making?

**Rationale 3.** Large corporations typically contract with health care providers to provide workplace wellness incentive programs. These wellness programs may include gym membership incentives, weight management, smoking cessation, personal finance and health and nutrition classes [24]. These opportunities are mostly offered to the most educated among us, as they make up a higher population of the workforce in these large organizations. These individuals also have more opportunities to work remotely. This makes exposure to the Coronavirus less likely for these individuals. As a result, the infection rates of those who are in less educated populations may be different from more educated populations.

## Study participants and context

The participants of this study were 26 rising ninth-grade students (14–15-years-old; 16 girls, 10 boys) who participated in an early college high school program operated by the College Planning Center at a state university in the northeastern United States. The goal of the early college program is to enable marginalized students in grades 9–12 to earn at least 12 transferable college credits, up to an associate degree, by the time they graduate from high school. The early college high school program combined rigorous college-level coursework with a high level of support and encouragement from teachers, counselors, mentors, and university professors who teach in the program. The students are referred to the program by their teachers from the public-school system, and they are admitted in the summer before their ninth-grade year.

The students' racial breakdown is 19% Asian, 8% Black, 54% Hispanic, and 19% White. These students were from five public schools across five school districts located within 20 miles of the university. Additionally, 42% of the students received free and reduced lunch, and 39% were classified as potential first-generation college students.

## Data generation and implementation process

The instructional unit was taught over three weeks via Zoom due to requirements for remote learning in the summer of 2020. The students attended classes twice per week that lasted for one hour. Students' written artifacts were collected each day and analyzed.

Below is a description of the day's events:

**Day 1:** Focus was to determine what the students already know about COVID-19, and to provide them with some of the guidelines provided by Center for Disease Control (CDC) [25]. Therefore, the students were asked:

1. KWL on COVID-19 [Students complete K&W portions of the graphic organizer. L portion will be completed at the end of the unit]

2. The students explored the CDC website site (https://www.cdc.gov/coronavirus/2019-ncov/downloads/2019-ncov-factsheet.pdf), read through the quick facts on COVID-19 and made notes on what stood out to them.

3. The students then engaged in a whole class discussion that were guided by the following questions:

    a. What concerns do you have about the CDC facts on Coronavirus?

    b. Do you agree with the suggested ways the CDC suggested to prevent Coronavirus?

    c. What, if anything, concerns you about these suggestions?

    d. Who is responsible for getting rid of Coronavirus?

    e. How do you propose this should be done?

**Day 2**: The Focus was to allow the students to think about biases in the COVID-19 discussion as they think about what relationships if any, exist between affluent communities, marginalized communities, and educational attainments.

1. The students watched the Trevor Noah video [26] (https://www.youtube.com/watch?v=NAh4uS4f78o).

2. After watching the video, answer the following:

    a. What did you See?

 b.  What do you Think?

 c.  What do you Wonder?

3.  The students were placed into small teams of 3 and instructed to use the U.S Census link below to gather information on their assigned cities and towns:
Research and record the Per-capita income [27] for their assigned cities and towns in the table
https://www.census.gov/quickfacts/fact/table/millburytownworcestercountymassachusetts, MA/PST045219

4.  Students use the link below to research and record the educational attainment [28] of their assigned cities or towns https://matracking.ehs.state.ma.us/

**Day 3:** The focus was to allow the students to start thinking about what relationships if any, exist between affluent communities, marginalized communities, educational attainment, and COVID– 19 infection rates [29]

1.  The students used the link below to research and record the COVID– 19 count per/100k and the rate per/100k for your assigned cities or towns for the identified between January 1st, 2020, to July 1, 2020 (available data up until time the students were actively in class).

https://www.mass.gov/info-details/covid-19-response-reporting#covid-19-cases-by-city/town-

**Day 4:** The focus of day 4 was on data analysis

1.  The students created a line graph of COVID-19 infections rates versus per-capita income for their assigned cities and towns

2.  The students created a line graph of COVID-19 infection rates versus educational attainment of their assigned cities and towns. The students answered each question based on Graph 1.

 a.  What pattern do you observe?

 b.  How do you feel about the curves you have observed in the graph?

 c.  What does this mean for your attitude towards science?

The students answered each question based on Graph 2.

a.  What pattern do you observe?

b.  How do you feel about the curves you have observed in the graph?

c.  What does this mean for your attitude towards science?

**Day 5:** The focus of day 5 was on interpretation the data analyzed to respond to the following: The students used the tables and graphs constructed to answer the following questions:

1.  What pattern, if any, do you notice?

2.  How would you describe the pattern(s) observed?

3.  What are your thoughts on the pattern(s) observed?

4.  What relationships, if any, do you believe exist between the pattern(s) observed and the rate at which COVID– 19 spreads in your city?

5. What relationships, if any, do you believe exist between the pattern(s) observed and the number of COVID– 19 cases in your city?

6. What relationships, if any, exist between the educational levels and COVID– 19 in your assigned cities and towns?

7. What relationships, if any, exist between the per capita income of your assigned cities and towns, and the rate and count of COVID– 19?

8. What, if anything, is concerning about the data gathered and the patterns observed?

9. What questions do you have?

10. How does the data you collect make you feel? Please explain.

## Data analysis-use critical thinking tenets from diversity work

Scientists' knowledge of COVID-19 at the time of this study was emerging, although we have gotten to know a lot more about this virus. As a result, the determination was made to start the instructional unit with a KWL chart. The KWL chart enables students to identify and activate prior knowledge, set learning goals, and identify new knowledge learned [30]. Including a KWL chart at the start of the study enabled the students identify what they already know about COVID-19, determine the relationships between the virus and a community's underlying health, wealth, and educational attainment, and use knowledge they would gain from the class to appeal to their elected officials and others to initiate change. The students' responses to the formative assessments in the instructional unit were used to identify relationships between wealth, educational attainment, and the impact of the COVID-19 Pandemic on their communities. The summative assessment required students to either write a letter to their town mayor to explain the significance of COVID-19 impact on their families, or create a public health information brochure to communicate to the others a problem health problem that the pandemic has exposed, andpotential actions that are needed to solve this problem. The summative assessment was used to assess the students' ability to use the knowledge gained to advocate for changes.

A constant comparative method of analysis [31] on the students' responses to the questions and associated activities were conducted to identify emergent themes. The author and two post-baccalaureate teacher candidates with qualitative data analysis experience were involved in the data analysis process. The data analysis process was divided into three phases. The first phase of the analysis examined themes apparent in artifacts of the students' work during days 1–3 of the unit. We identified segments of the data from these three days that included information on the students' prior knowledge about COVID-19, students' understanding of biases in news media reports on COVID-19, and the relationships between affluent communities, marginalized communities, educational attainment, and these communities' COVID-19 infection rates.

In the second phase of the analysis, we examined the students' artifacts from days 4 & 5 to identify emergent themes generated from their data analyses of wealth, health, and educational attainments on the COVID-19 infection rates from their assigned Massachusetts communities. Our analysis allowed us to determine whether the students were able to determine who is impacted the most by the COVID-19 Pandemic, what are some potential reasons for being impacted, and to share their feelings.

In the third phase of the analysis, we examined the students' abilities to explain why communities in Massachusetts are impacted differently by the COVID-19 Pandemic. The final phase of the analysis allowed us to elaborate on the students' abilities to think critically about steps that needed to be taken to reduce the impact of COVID-19 on themselves and their communities.

## Results and discussion

The findings demonstrated how ninth-graders' understanding of the relationships between COVID-19 on health, wealth, and educational attainments drive their decisions to advocate for changes to protect their community. Details below are the three main areas of findings that are aligned to the two research questions below:

1. *What relationships, if any, exist between health in Massachusetts communities and the COVID-19 infection rates?*

This study took place during summer 2020, the height of the COVID-19 Pandemic. At the time of this study, the COVID-19 vaccines were not yet developed, and scientists' knowledge of the virus was still evolving, even more so than now. The country was also still partially in lockdown mode. As it relates to what the students knew about COVID-19, their source of this knowledge was reports from news media, CDC bulletin, and their lived experiences. When asked to state what they know about COVID-19, the students' responses were categorized into four main themes (what students knew about COVID-19; Table 1).

The students' responses in the table indicate that they have a relatively nuanced understanding of what COVID-19 is, its impact on the population, and who is impacted the most by the virus. The students stated that they knew that the virus was spreading among marginalized populations. The students also reasoned that the elderly are more susceptible to the virus because the elderly are expected to have more underlying health problems. The students' knowledge of the virus seems to be more informed as opposed to many of our elected officials and media personalities [26]. Whereas the students knew that the virus is a real threat to the lives of many, some American politicians and radio personalities saw the virus as a hoax.

When asked to state what they wanted to know about COVID-19, the students' responses' data identified several exciting themes (what I want to know about COVID-19; Table 2).

Interestingly, much of what the students wanted to know about COVID-19 has since become common knowledge as the virus rages locally, nationally, and internationally. We have seen the virus killing more individuals from minoritized groups. We have also learned more about complications due to COVID-19. Since then, mutations from the many variants of the virus have wreaked havoc on people's lives. The Delta strain of the virus has killed millions worldwide, and we are all now worried about the new Omicron strains of the virus. Also, we now have vaccines that vary in their degree of protection against the different strains of the virus. The students wanted to know if the virus would become like the seasonal flu, and we seem to be heading in that direction. The students wanted to know more than many of our

**Table 1. What the students knew about COVID-19.**

| Themes | What I know | Researchers' Interpretation |
|---|---|---|
| • COVID-19 impact on Millions | • Covid-19 is a virus that has affected millions of people. | The students understand that the virus is real and is having adverse health effects on people around the world |
| • Respiratory disease | • It's a disease that attacks your lungs | The students understood that the Coronavirus is a respiratory virus that affects the lungs |
| • Affects marginalized population more | • COVID-19 spreads more areas where people are less fortunate. | The students knew that the virus was impacting marginalized populations much more than others. |
| • Affects elderly and people with underlying medical conditions more | • Mainly affects older people and people with underlying medical conditions | The students knew that the virus was impacting the elderly and people with underlying health conditions more than anyone else. |

**Table 2. What the students wanted to know about COVID-19.**

| Themes | What I want to know | Researchers' Interpretation |
|---|---|---|
| • COVID-19 doesn't impact all people the same | • Why are some people more at risk? Why COVID-19 does not affect younger people as much as older people and people with underlying condition. | The students heard that COVID-19 doesn't affect all people the same way. They observed that many are able to recover after contracting the virus, while others die. As a result, they wanted to learn why some people are more susceptible to the virus than others. For example, they wanted to know why young people were more likely to survive if they contracted the virus than the elderly with an underlying condition(s). |
| • Plans to eliminate the virus | • How are we planning to solve this pandemic? | The students were very concerned about the plans to eliminate the virus. They understood that vaccines will help, but at the time there were none available. They wanted to know what plans the government had in place to get rid of the virus. |
| • Mutation | • Is Covid-19 going to be a seasonal thing like the Flu?<br>• Is there going to be another stage of the COVID-19 virus that is going to kill many people again? | The students used what they learned about mutation in their science courses to inquire about the possibility of the virus becoming like the seasonal flu. They wanted to know if the virus could mutate to a new strain and the impact this would have on people's lives. |
| • Vaccine Availability | • Will there be a vaccine that is going to work to help COVID-19? | The students wanted to know if scientists would be able to create an effective vaccine. |

elected officials wanted to know. As the virus continues to sicken and kill many Americans, elected officials make erroneous claims, including, but not limited to statements such as "real America is done with COVID" [32]. The students in this investigation understood the seriousness of the virus and wanted answers to their questions. They wanted to know when we will be able to control the virus. The students also wanted to know why the virus affects some people more than others. While the students in this study were asking these questions, many elected officials from the local, regional, and national levels were calling on the general public to be cautious about COVID-19. At the same time, some elected officials, radio personalities, and others were touting the virus as a hoax. Because of this mixed messaging, the author wanted to provide the students with the known facts about COVID-19 at the time. Therefore, the determination was made to have the students review the Center for Disease Control and prevention guidelines on the known facts about COVID-19 (see S1). At time of this study, there were no available vaccines (facts about COVID-19; Table 3).

The students' responses indicated that they know the virus was for real and that it could be potentially deadly if one should contract the virus. The students learned about the symptoms and how the virus spreads. They also learned that anyone could contract the virus as well as how to how to protect themselves against the virus. This information was critical information for the students to have especially since the communities in which they live were experiencing very high COVID-19 positivity rates.

To get the students to think more carefully about the COVID-19 guidelines advertised by the Center for Disease Control and Prevention, and to help them formulate their own opinions on the guidelines, they were asked to respond to the following questions (COVID-19 fact sheet questions; Table 4).

The first question sought to determine the students' concerns about the CDC's facts advertised.

**Table 3. Students' statements about COVID-19 after reviewing the CDC's guidelines.**

| Know about COVID-19 | • Covid-19 is caused by a virus that is contagious.<br>• The virus has spread across the world at a rapid rate.<br>• Symptoms for the disease could be anywhere from mild flu-like symptoms or something more severe. |
| --- | --- |
| Know how COVID-19 is spread | • Coming within 6 feet of a person with the virus is a way of spreading it.<br>• You could get infected from respiratory droplets the person releases when they talk, cough, or sneeze.<br>• You could also get the virus by touching an object or surface that has the virus on it. |
| Protect yourself and others from COVID-19 | • Since there is no vaccine, the best way to protect yourself is by not coming in contact with sick persons.<br>• Stay at home and avoid standing too close to people.<br>• Wear a mask or cloth face covering in public.<br>• Disinfect surfaces frequently<br>• Wash your hands often and use hand sanitizer that contains at least 60% alcohol. |
| Know Symptoms | • Symptoms include coughing, fever, difficulty breathing, headache, muscle ache, loss of taste or smell, sore throat, nausea, vomiting, congestion, runny nose, diarrhea, and fatigue. |
| Know Treatments | • There are no treatments as of now, but you can get some medical care to relieve your symptoms |
| Know your risk for severe illness | • Anyone can get it.<br>• People of older age, with type A blood, and with any underlying respiratory illnesses or medical conditions may be at a higher risk |

While the students were naturally concerned for the well-being of their friends and loved ones with pre-existing conditions contracting the virus, they were still skeptical that the information provided was accurate. The students' skepticism about the scientific facts on COVID-19 outlined by the CDC is critical because scientific skepticism is rising [33–35]. Studies have reported that the rise in science skepticism is more aligned to climate change, vaccination, and genetic modification [36]. It is worth noting that skepticism in all these domains might have far reaching effects on our way of life, as evident from the COVID-19 fallout. Almost two years after the virus was first detected, and over one million deaths in the United States, and millions of COVID-19 related deaths worldwide, many Americans are still hesitant to get their COVID-19 vaccines. As vaccine hesitancy grows, people are still dying by the thousands daily around the world. Most of these deaths are as a result of COVID-19 complications among the unvaccinated population. This skepticism that is driving COVID-19 vaccine hesitancy will have far reaching effects on our way of life moving forward.

If the science education community is unable to teach the population in general, and students in particular, how to critically evaluate evidence, then this continuous rise in scientific skepticism has the potential to result in catastrophic effects on public health, the economy, and the environment [37,38].

While the students were skeptical of the CDC's reported facts on Coronavirus, the students agreed with the CDC's evidence that COVID-19 is an illness caused by a virus that can spread from person to person. They also agreed with the social distancing guidelines presented and the methods to protect oneself from the virus. However, because of their family living situation, they acknowledge that they will not always have enough space in their house to separate from loved ones or even pets who could potentially be infected with the virus. They understand the need to social distance from others who could potentially be infected with the virus, but they believed they themselves would be infected should one of the household members becomes affected because they didn't have enough space in their homes to isolate. The students

**Table 4. Students' reactions to the CDC's April 2020 COVID-19 fact sheet.**

| Questions | Students' Response | Themes | Researchers" Interpretation |
|---|---|---|---|
| What concerns do you have about the CDC facts on Coronavirus? | I have concerns about people I know with preexisting conditions that means they might be more at risk for the disease. | Concerns for well-being of people with pre-existing conditions based on CDC facts on COVID-19 | The students were able to make the connection between the virus and the potential impact that it would have on family members with pre-existing conditions. |
| | Because this disease is so new maybe they don't have enough facts because we are learning new things about Covid-19 everyday maybe they might put out something that is inaccurate that leads people into thinking something that is far from the truth | Skepticism about the CDC facts on COVID-19 | The students stated that COVID-19 is new to everyone, so perhaps the CDC doesn't have all of the facts, and so some of the information from the CDC facts sheet might be inaccurate. |
| Do you agree with the ways the CDC suggested to prevent Coronavirus? | I agree with the suggested guidelines to prevent the disease from spreading. There is a lot of scientific evidence supporting the CDC's ways to prevent Coronavirus. | Observed scientific evidence in the CDC facts on COVID-19 | The students agreed with the CDC's evidence that COVID-19 is an illness caused by a virus that can spread from person to person. They also agreed with the social distancing guidelines presented and the methods to protect oneself from the virus. |
| | Mostly, I do agree with the suggested ways the CDC suggested to prevent the Coronavirus. There are some facts that I did not agree with including the one about separating yourself from pets and people in your own home. I think that this something that is difficult to do, and something that I did not know we were supposed to be doing. Some of the ways such as staying at home if you're sick, and to get medical care if needed, I do agree with. The symptoms for COVID-19 can be bad, and the virus can hurt your respiratory system, so it is important to seek medical attention if needed. | Agreed with the CDC's guidelines on preventing the spread of COVID-19, but admit to not agree in separating oneself from pets and other family members | The students understand that the virus posed significant risks to one's health if contracted. However, they questioned the feasibility for households with fewer resources to isolate from each other. They also raised question on how possible it is for these household to even isolate from pets. |
| What if anything concerns you about these suggestions? | I would not be concerned about much except the mental health portion as that can affect most people on this world. | Not concerned about the CDC's guidelines, but wanted to see more advice on mental measures to cope with the virus and its effects on people | The students were worried and scared for the well-being of their loved ones, especially those with preexisting health conditions. As a result, they wanted to see more information from the CDC on mental health coping strategies. |
| | I don't like the fact that we have to wear face masks and because of social distancing we can't hug or high five anyone, but I do understand that there are certain sacrifices we have to make whether we like it, which will just help society in the long run, and there's usually light at the end of every tunnel. | Not too thrilled with the CDC's face masks guidelines, but willing to wear masks when around others | The students were not thrilled about wearing face masks and social distancing from friends and loved ones who are not in the same household. However, they do understand that sacrifices are needed to protect themselves and others. They believe that if we all make these sacrifices of wearing face masks in public and practice social distancing, then then society will beat the virus. |
| Who is responsible for getting rid of Coronavirus? How do you proposed this should be done? | Society is responsible for getting rid of Coronavirus because the virus spreads among humans, so everyone plays a role in helping to get rid of the virus. The way to do this is by following social distancing guidelines, wearing masks, and limiting in-person contact as much as possible until scientists have figured out some kind of vaccine so that we can all go back to what we used to be able to do and considered to be "normal." | Getting rid of the virus is the responsibility of all | The students understand that controlling the Coronavirus is the responsibility of all. They believe that we all have a part to play in combating the virus. |

were very concerned about their safety and their loved ones' safety since many of their parents still had to work on the frontline. They were concerned that this created more opportunities for them and their loved ones to be infected with the virus.

The students raised questions about why minoritized population and the elderly were more susceptible to COVID-19. The students were able to associate age with underlying health conditions as they reported that they would expect the elderly to have more health issues in comparison to the younger generation. However, the students did not seek answers to why, nor did they associate long-term exposure to environmental toxins to ill health among the elderly population, especially the elderly population from minoritized groups.

2. *What relationships, if any, exist between wealthy communities in Massachusetts and COVID-19 infection rates?*

To generate a discussion among the students on potential biases in news reports on the COVID-19, they were asked to watch the Trevor Noah, April 3, 2020, The Daily Show monologue [26]. Before watching the video, the students were introduced to the See-Think-Wonder method [39]. This was done to allow the students to slow down their thinking and to carefully observe before drawing conclusions. The students overwhelmingly stated that they saw people not telling the truth about the virus. The students stated that they think people should be responsible for what they say on the airways, and they wonder why elected officials and media personalities would try to mislead the public about something that is so serious. The students then engaged in their own research to determine the COVID-19 infection rates across Massachusetts. The students were assigned specific cities and towns to investigate (cities and towns, Table 5).

They investigated the per-capita income [27] for their assigned cities and towns, the educational attainment [28], the infection count/100K, and the infection rates per 100K [29] between January 1-April 14, 2020, to January 1-July 1, 2020 (latest data available at the time of the study).

Among all the groups, the common cities and towns were Dover, Massachusetts, and Framingham, Massachusetts. Dover was assigned to all the groups because it was the wealthiest. Framingham was assigned because of the students' connection to the city.

Due to space limitations, the Table 6A–6D, and Figs 1 and 2 in the body of the report detail the findings from group 1. S1 File shows detailed tables for groups 2–7. S2 File shows detailed figures of groups 2–7.

The students were then asked to use the data collected to create line graphs of the COVID-19 infection rates versus the per-capita income. The students were asked to identify patterns observed in the graphs, and to state how the data they collected and analyzed make them feel (patterns, Fig 1).

1. What pattern, if any, do you observe in graph 1?
   *A pattern I observe is the cities/towns with lower per-capita income, have higher infection*

**Table 5. Assigned Massachusetts cities and towns investigated.**

| Group 1 | Group 2 | Group 3 | Group 4 | Group 5 | Group 6 | Group 7 |
|---|---|---|---|---|---|---|
| Dover | Dover | Dover | Dover | Dover | Dover | Dover |
| Sudbury | Wellesley | Concord | Norwell | Newton | Carlisle | Medfield |
| Westwood | Boxford | Brookline | Weston | Needham | Sherborn | Lincoln |
| Everett | Lowell | New Bedford | Worcester | Chelsea | Southbridge | Boston |
| Fall River | Fitchburg | Lawrence | Brockton | Holyoke | Springfield | Milford |
| Framingham | Framingham | Framingham | Framingham | Framingham | Framingham | Framingham |
| **State** | **State** | **State** | **State** | **State** | **State** | **State** |

**Table 6. a. Group 1 per-capita incomes, educational attainment, and COVID-19 infection rates between January 1, 2020-April 29, 2020.** b. Group 1 per-capita incomes, educational attainment, and COVID-19 infection rates between January 1, 2020-May 20, 2020. c. Group 1 per-capita incomes, educational attainment, and COVID-19 infection rates between January 1, 2020-June 10, 2020. d. Group 1 per-capita incomes, educational attainment, and COVID-19 infection rates between January 1, 2020-July 1, 2020.

| Town | Per-capita Income | Educational Attainment in 2017 | | | | | COVID-19 January 1-April 14, 2020 | | COVID-19 January 1-April 22, 2020 | | COVID-19 January 1-April 29, 2020 | |
|---|---|---|---|---|---|---|---|---|---|---|---|---|
| | | No HS Dip. | HS Dip/GED | Some Coll. | AS or BA | Adv. Deg. | Count/100k | Rate/100k | Count/100k | Rate/100k | Count/100k | Rate/100k |
| Dover | $115,686 | 2% | 5% | 6% | 40% | 47% | 14 | 268.65 | 14 | 268.65 | 14 | 268.65 |
| Sudbury | $75,699 | 1% | 8% | 9% | 41% | 41% | 34 | 189.90 | 55 | 307.20 | 76 | 424.49 |
| Westwood | $62,949 | 3% | 13% | 9% | 40% | 35% | 48 | 325.66 | 67 | 454.56 | 82 | 556.33 |
| Everett | $26,591 | 19% | 37% | 20% | 18% | 5% | 444 | 914.78 | 716 | 1475.18 | 1004 | 2068.55 |
| Fall River | $21,257 | 28% | 35% | 16% | 18% | 5% | 150 | 167.70 | 191 | 213.53 | 481 | 537.75 |
| Framingham | $33,695 | 11% | 23% | 13% | 33% | 20% | 263 | 353.18 | 438 | 588.19 | 816 | 1095.80 |
| **State** | **$41,794** | **9.3%** | **23.3%** | **15.3%** | **32%** | **20.10%** | **28,163** | **487.76** | **42,944** | **616.41** | **60,265** | **865.03** |

| Town | Per-capita Income | Educational Attainment in 2017 | | | | | COVID-19 January 1-May 6, 2020 | | COVID-19 January 1-May 13, 2020 | | COVID-19 January 1-May 20, 2020 | |
|---|---|---|---|---|---|---|---|---|---|---|---|---|
| | | No HS Dip. | HS Dip/GED | Some Coll. | AS or BA | Adv. Deg. | Count/100k | Rate/100k | Count/100k | Rate/100k | Count/100k | Rate/100k |
| Dover | $115,686 | 2% | 5% | 6% | 40% | 47% | 14 | 268.65 | 16 | 307.03 | 17 | 326.22 |
| Sudbury | $75,699 | 1% | 8% | 9% | 41% | 41% | 98 | 547.37 | 163 | 910.42 | 174 | 971.86 |
| Westwood | $62,949 | 3% | 13% | 9% | 40% | 35% | 93 | 630.96 | 99 | 671.67 | 112 | 759.87 |
| Everett | $26,591 | 19% | 37% | 20% | 18% | 5% | 1212 | 2497.10 | 1366 | 2814.38 | 1474 | 3036.90 |
| Fall River | $21,257 | 28% | 35% | 16% | 18% | 5% | 652 | 728.92 | 868 | 970.41 | 1051 | 1175.00 |
| Framingham | $33,695 | 11% | 23% | 13% | 33% | 20% | 1159 | 1556.41 | 1347 | 1808.87 | 1493 | 2004.93 |
| **State** | **$41,794** | **9.3%** | **23.3%** | **15.3%** | **32%** | **20.10%** | **72025** | **1033.83** | **80,497** | **1155.44** | **88970** | **1277.06** |

| Town | Per-capita Income | Educational Attainment in 2017 | | | | | COVID-19 January 1-May 27, 2020 | | COVID-19 January 1-June 3, 2020 | | COVID-19 January 1-June 10, 2020 | |
|---|---|---|---|---|---|---|---|---|---|---|---|---|
| | | No HS Dip. | HS Dip/GED | Some Coll. | AS or BA | Adv. Deg. | Count/100k | Rate/100k | Count/100k | Rate/100k | Count/100k | Rate/100k |
| Dover | $115,686 | 2% | 5% | 6% | 40% | 47% | 18 | 345.41 | 18 | 345.41 | 18 | 345.41 |
| Sudbury | $75,699 | 1% | 8% | 9% | 41% | 41% | 181 | 1010.95 | 188 | 1050.05 | 191 | 1066.81 |
| Westwood | $62,949 | 3% | 13% | 9% | 40% | 35% | 120 | 814.15 | 122 | 827.72 | 122 | 827.72 |
| Everett | $26,591 | 19% | 37% | 20% | 18% | 5% | 1565 | 3224.38 | 1647 | 3393.33 | 1692 | 3486.04 |
| Fall River | $21,257 | 28% | 35% | 16% | 18% | 5% | 1218 | 1361.70 | 1363 | 1523.80 | 1472 | 1645.66 |
| Framingham | $33,695 | 11% | 23% | 13% | 33% | 20% | 1625 | 2182.20 | 1681 | 2257.40 | 1700 | 2282.91 |
| **State** | **$41,794** | **9.3%** | **23.3%** | **15.3%** | **32%** | **20.10%** | **94220** | **1352.42** | **97964** | **1406.16** | **100158** | **1437.65** |

| Town | Per-capita Income | Educational Attainment in 2017 | | | | | COVID-19 January 1-June 17, 2020 | | COVID-19 January 1-June 24, 2020 | | COVID-19 January 1-July 1, 2020 | |
|---|---|---|---|---|---|---|---|---|---|---|---|---|
| | | No HS Dip. | HS Dip/GED | Some Coll. | AS or BA | Adv. Deg. | Count/100k | Rate/100k | Count/100k | Rate/100k | Count/100k | Rate/100k |
| Dover | $115,686 | 2% | 5% | 6% | 40% | 47% | 18 | 345.41 | 18 | 345.41 | 18 | 345.41 |
| Sudbury | $75,699 | 1% | 8% | 9% | 41% | 41% | 194 | 1083.56 | 197 | 1100.32 | 195 | 1089.15 |
| Westwood | $62,949 | 3% | 13% | 9% | 40% | 35% | 121 | 820.93 | 126 | 854.85 | 127 | 861.64 |
| Everett | $26,591 | 19% | 37% | 20% | 18% | 5% | 1724 | 3551.97 | 1747 | 3599.36 | 1765 | 3636.45 |
| Fall River | $21,257 | 28% | 35% | 16% | 18% | 5% | 1543 | 1725.04 | 1566 | 1750.75 | 1595 | 1783.18 |

*(Continued)*

**Table 6.** (Continued)

| | | | | | | | | | | | |
|---|---|---|---|---|---|---|---|---|---|---|---|
| Framingham | $33,695 | 11% | 23% | 13% | 33% | 20% | 1707 | 2292.31 | 1720 | 2309.77 | 1734 | 2328.57 |
| **State** | **$41,794** | **9.3%** | **23.3%** | **15.3%** | **32%** | **20.10%** | **101654** | **1459.12** | **102762** | **1475.03** | **103858** | **1490.76** |

*rates, and the infection rates increased over time. Another pattern I observe is the cities/towns with higher per-capita income, have lower infection rates and have either stayed the same or very little increased infection rates.*

2. How does the data you collect make you feel? Please explain. The response is typical of what the students wrote:
*The data I have collected makes me feel sympathetic. According to my research, many less fortunate people have a higher chance of getting infected. Further spreading the virus and making their lives harder than in already is.*

3. What does this mean for your attitude towards science?
*It helps me see that science helps prove many things and is very reassuring during these times of panic.*

The above response identified that cities and towns in Massachusetts with a higher per-capita income had a lower infection rate. This caused the students to feel sympathetic towards those people who were being infected with the virus. In addition, because of the skepticism in conservative media about the science behind COVID-19, it was reassuring to see that this simple study impacted the students to the point where they felt comforted during the pandemic.

3. *What relationships, if any, exist between educational attainments and COVID-19 infection rates in Massachusetts cities and towns?*

The students were asked to plot educational attainment versus infection rates for their assigned cities and towns (comparison between COVID-19 infection rates and educational attainments for the students who were assigned to group 1; Fig 2). See S2 File for groups 2–7 graphs.

1. What pattern, if any, do you observe in graph 2?
*The more educated the city the less the rate of coronavirus in the city*

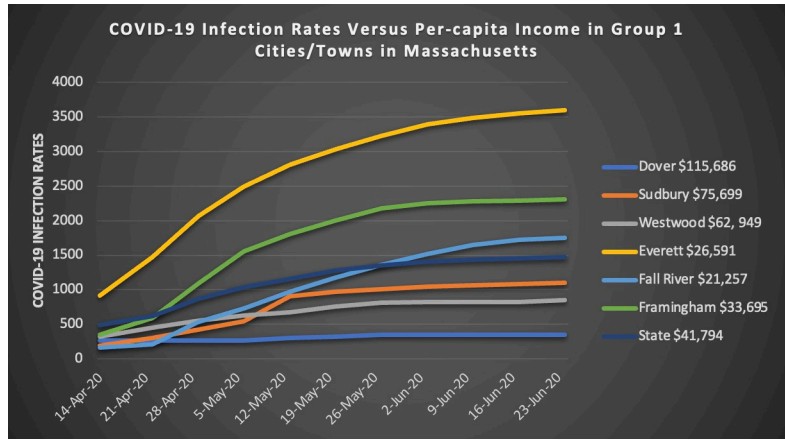

**Fig 1. Group 1.** COVID-19 infections rates versus per-capita income.

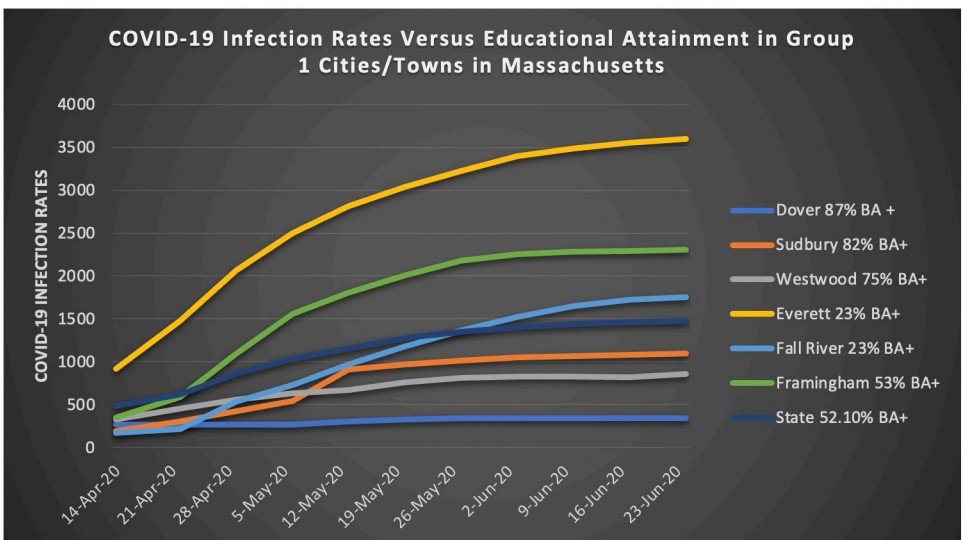

**Fig 2. Group 1.** COVID-19 infection rates versus educational attainment.

2. How does the data you collect make you feel? Please explain.

*After reading and analyzing all the data, creating the graphs, and answering the questions, I feel disappointed, sad, and confused at the same time. The Covid-19 was a virus that attacked countries from all over the world and infected millions of people, but by looking at the statistics of these cities and towns around Massachusetts, it clearly states that only the people who were better educated and more financially resourceful could protect themselves from this virus successfully, while the others took the worst and had a lot of people infected by the virus. So, this is how humanity solves its problems? Only the rich get the best and are safe while the poor suffer to survive, and people act like this is not an issue? It is in these instances that you realize that humanity is always predetermined by who has the most resources, which is something that has always been around throughout all human history. We are better than this, and we should do something about this, but due to only the rich also being the ones in power, I guess we will have to live like this for all of eternity.*

I feel that we should make it our mission as a society to become more educated as people need to learn more to stay healthy.

The students identified from the data they collected and analyzed that communities in Massachusetts with more financial resources were impacted less by the virus. When asked why this is so during discussion, the students identified that these communities were typically more educated which means residents from these communities had a better chance of working from home, which creates less opportunity to be around infected people.

The students expressed disappointment, sadness, and confusion as to why the educated and wealthy among us fare much better with COVID-19. They reiterated that our society is better than this and that we need to do more to help each other.

The students also stated that they sympathized with those who are less fortunate. Below is a typical response:

*It makes me mad, and sympathetic, because I see very fortunate people and very unfortunate people. And no one is doing anything about it we all have to work together if we want to beat*

*this and we can't do that without the help of the fortunate. And the numbers show more money equals fewer cases we need to even it out.*

## Conclusion

At the time of writing, it is now almost two years since Coronavirus was first detected in the United States. Since its detection, countries worldwide have initiated lockdowns, over one million fellow Americans have died, deaths due to COVID-19 complications are occurring daily, and scientists have been able to produce several vaccines to help combat the virus.

While the virus continues to kill thousands of Americans and others worldwide and cause daily mayhem to our way of life, vaccine hesitancy is at an all-time high. Many of the reasons given by vaccine skeptics are not aligned with the science behind vaccination and how vaccines work. For example, in a recent radio interview [40], the United States Senator wondered out loud to his audience and asked:

1. *Why do we assume that the body's natural immune system isn't the marvel that it is*?

2. *Why do we think that we can create something better than God in terms of combating disease*?

The senator seems to believe that public health officials should allow people to get COVID-19 because of the body's natural immunity to rid the body of the virus.

This suggestion seems counterproductive to us ridding ourselves of the virus. The senator's comments are also dangerous as we have seen the death and destruction that COVID-19 is having on communities in general and minoritized communities in particular. While one doesn't often see bible verses written in science education articles, it was difficult to gloss over the Senator's assertion of why scientists believe that they can create something better than God in terms of combatting disease. I would like to remind the senator of proverbs 1: verse 7, which reads, "The fear of the Lord is the beginning of knowledge: fools despise wisdom and instruction." The senator would agree that we are blessed with wisdom. The intent is for us to use our wisdom to make sound decisions.

Perhaps, the senator needs to take a second to read what King Solomon meant. With COVID-19 affecting the least among us more than anyone else, it is not time to stoke fear about taking a COVID-19 vaccine. The students' investigation has shown that COVID-19 infection rates are higher in communities in Massachusetts that are less healthy, poor, and least educated. The students also questioned whether this is how humanity should go about solving its problems. Let's educate students to ask pertinent questions that will enable them to make informed decisions.

## Supporting information

**S1 Fig. CDC.** COVID-19 guidelines.
(DOCX)

**S2 Fig.**
(TIF)

**S3 Fig.**
(TIF)

**S4 Fig.**
(TIF)

**S5 Fig.**
(TIF)

**S6 Fig.**
(TIF)

**S7 Fig.**
(TIF)

**S8 Fig.**
(TIF)

**S9 Fig.**
(TIF)

**S10 Fig.**
(TIF)

**S11 Fig.**
(TIF)

**S12 Fig.**
(TIF)

**S13 Fig.**
(TIF)

**S14 Fig.**
(TIF)

**S1 File. Detailed tables for groups 2–7.** Table 7a. Group 2 per-capita incomes, educational attainment, and COVID-19 infection rates between January 1, 2020-April 29, 2020. Table 7b. Group 2 per-capita incomes, educational attainment, and COVID-19 infection rates between January 1, 2020-May 20, 2020. Table 7c. Group 2 per-capita incomes, educational attainment, and COVID-19 infection rates between January 1, 2020-June 10, 2020. Table 7d. Group 2 per-capita incomes, educational attainment, and COVID-19 infection rates between January 1, 2020-July 1, 2020. Table 8a. Group 3 per-capita incomes, educational attainment, and COVID-19 infection rates between January 1, 2020-April 29, 2020. Table 8b. Group 3 per-capita incomes, educational attainment, and COVID-19 infection rates between January 1, 2020-May 20, 2020. Table 8c. Group 3 per-capita incomes, educational attainment, and COVID-19 infection rates between January 1, 2020-June 10, 2020. Table 8d. Group 3 per-capita incomes, educational attainment, and COVID-19 infection rates between January 1, 2020-July 1, 2020. Table 9a. Group 4 per-capita incomes, educational attainment, and COVID-19 infection rates between January 1, 2020-April 29, 2020. Table 9b. Group 4 per-capita incomes, educational attainment, and COVID-19 infection rates between January 1, 2020-May 20, 2020. Table 9c. Group 4 per-capita incomes, educational attainment, and COVID-19 infection rates between January 1, 2020-June 10, 2020. Table 9d. Group 4 per-capita incomes, educational attainment, and COVID-19 infection rates between January 1, 2020-July 1, 2020. Table 10a. Group 5 per-capita incomes, educational attainment, and COVID-19 infection rates between January 1, 2020-April 29, 2020. Table 10b. Group 5 per-capita incomes, educational attainment, and COVID-19 infection rates between January 1, 2020-May 20, 2020. Table 10c Group 5 per-capita incomes, educational attainment, and COVID-19 infections rate between January 1, 2020-June 10, 2020. Table 10d. Group 5 per-capita incomes, educational attainment, and COVID-19 infection rates between January 1, 2020-July 1, 2020. Table 11a. Group 6 per-capita incomes,

educational attainment, and COVID-19 infection rates between January 1, 2020-April 29, 2020. Table 11b. Group 6 per-capita incomes, educational attainment, and COVID-19 infection rates between January 1, 2020-May 20, 2020. Table 11c. Group 6 per-capita incomes, educational attainment, and COVID-19 infection rates between January 1, 2020-June 10, 2020. Table 11d. Group 6 per-capita incomes, educational attainment, and COVID-19 infection rates between January 1, 2020-July 1, 2020. Table 12a. Group 7 per-capita incomes, educational attainment, and COVID-19 infection rates between January 1, 2020-April 29, 2020. Table 12b. Group 7 per-capita incomes, educational attainment, and COVID-19 infection rates between January 1, 2020-May 20, 2020. Table 12c. Group 7 per-capita incomes, educational attainment, and COVID-19 infection rates between January 1, 2020-June 10, 2020. Table 12d. Group 7 per-capita incomes, educational attainment, and COVID-19 infection rates between January 1, 2020-July 1, 2020.
(DOCX)

**S2 File. Detailed figures of groups 2–7.** Fig 3. Group 2. Per-capita income. Fig 4. Group 2. Educational attainment. Fig 5. Group 3. Per-capita income. Fig 6. Group 3. Educational attainment. Fig 7. Group 4. Per-capita income. Fig 8. Group 4. Educational attainment. Fig 9. Group 5. Per-capita income. Fig 10. Group 5. Educational attainment. Fig 11. Group 6. Per-capita income. Fig 12. Group 6. Educational attainment. Fig 13. Group 7. Per-capita income. Fig 14. Group 7. Educational attainment.
(DOCX)

**S3 File. Minimum data set.**
(DOCX)

## Acknowledgments

I thank all the pupils for participating in this study. Your willingness to participate is greatly appreciated.

## Author Contributions

**Conceptualization:** Wardell Powell.

**Data curation:** Wardell Powell.

**Formal analysis:** Wardell Powell.

**Funding acquisition:** Wardell Powell.

**Investigation:** Wardell Powell.

**Methodology:** Wardell Powell.

**Project administration:** Wardell Powell.

**Resources:** Wardell Powell.

**Supervision:** Wardell Powell.

**Validation:** Wardell Powell.

**Visualization:** Wardell Powell.

**Writing – original draft:** Wardell Powell.

**Writing – review & editing:** Wardell Powell.

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
