## [Decision Letter · Decision Letter 0]

10 Oct 2022

PONE-D-22-19529A socioscientific issues approach to ninth-graders’ understanding of COVID-19 on health, wealth, and educational attainmentsPLOS ONE

Dear Dr. Powell,

Thank you for submitting your manuscript to PLOS ONE. After careful consideration, we feel that it has merit but does not fully meet PLOS ONE’s publication criteria as it currently stands. Therefore, we invite you to submit a revised version of the manuscript that addresses the points raised during the review process.

Please submit your revised manuscript by Nov 24 2022 11:59PM If you will need more time than this to complete your revisions, please reply to this message or contact the journal office at plosone@plos.org. Please include the following items when submitting your revised manuscript:A rebuttal letter that responds to each point raised by the academic editor and reviewer(s). You should upload this letter as a separate file labeled 'Response to Reviewers'.A marked-up copy of your manuscript that highlights changes made to the original version. You should upload this as a separate file labeled 'Revised Manuscript with Track Changes'.An unmarked version of your revised paper without tracked changes. You should upload this as a separate file labeled 'Manuscript'.

We look forward to receiving your revised manuscript.

Kind regards,

Ana Larissa Gomes Machado, Ph.D

Academic Editor

PLOS ONE

Journal Requirements:

Additional Editor Comments:

2. Please change "female” or "male" to "woman” or "man" as appropriate, when used as a noun (see for instance https://apastyle.apa.org/style-grammar-guidelines/bias-free-language/gender)."

“The funders had no role in study design, data collection and analysis, decision to publish, or preparation of the manuscript”

6. Please amend your manuscript to include your abstract after the title page.

Dear Dr Powell

I am writing today because your manuscript “A socioscientific issues approach to ninth-graders’ understanding of COVID-19 on health, wealth, and educational attainments", which you submitted to PLOS ONE, has been reviewed. Please find the reviewer comments at the bottom of this letter.

I regret to inform you that the reviewers have raised serious concerns, and therefore your paper cannot be accepted for publication at this time. However, since the reviewers do find merit in the paper, we would be willing to reconsider if you wish to undertake major revisions and resubmit, fully addressing the referees’ concerns enumerated below.

Please note that resubmitting your manuscript does not guarantee eventual acceptance. Should you choose to do so, the manuscript will be subject to re-review before a decision is rendered.

If you chose to revise your manuscript, please highlight the changes you make in the manuscript by using the track changes mode in MS Word or by using bold or colored text. Please also upload a file that details your responses to the comments made by the reviewers. Please be as specific as possible in your response to the reviewers but do not include any author contact information and/or names as this will be shared with the reviewers and it is important to keep the review process anonymous.

Reviewers' comments:

Reviewer's Responses to Questions

**Comments to the Author**

1. Is the manuscript technically sound, and do the data support the conclusions?

Reviewer #1: Partly

Reviewer #2: Yes

2. Has the statistical analysis been performed appropriately and rigorously? 

Reviewer #1: N/A

Reviewer #2: Yes

3. Have the authors made all data underlying the findings in their manuscript fully available?

Reviewer #1: Yes

Reviewer #2: Yes

4. Is the manuscript presented in an intelligible fashion and written in standard English?

Reviewer #1: Yes

Reviewer #2: Yes

5. Review Comments to the Author

Reviewer #1: In my opinion the research questions are out of alignment with the main story of the paper. The research questions seem to be what the students are studying however I think the real story is what the students were thinking about, and their proficiency in making reasoned claims with evidence using the data provided. Also, did this exercise help the students understand the need for using data to inform decision-making?

There is a good amount of literature on a justice orientation in science education that can be included as part of the theoretical framework. The author provides a good background and placing the pandemic in context, as well as identifying the need for a study by highlighting the skepticism and questionable claims made by public figures.

I would encourage the author to reconsider the main message from the study. In my mind the study should focus on the students’ perspectives and their thoughts rather than determining the relationship between the indicators present.

Reviewer #2: - Sampling technique is required along with the actual size of sample.

- this paper is more of qualitative research realm; so in results, discussion the data collected from focused group interview needs to be given more emphasis than document analysis.

- the real-life situations and challenges of the sample need to be highlighted more; this would be a better ground to give suggestive measures in conclusion part.

- the paper is nicely drafted.

---

## [Author Response · Author response to Decision Letter 0]

22 Dec 2022

Academic Editor

PLOS ONE

November 29, 2022

Dear Editor,

I am pleased to have been allowed to revise my manuscript entitled "A socioscientific issues approach to ninth-graders' understanding of COVID-19 on health, wealth, and educational attainments" (PONE-D-22-19529). In the revised manuscript, I carefully considered the reviewers' comments and have succinctly made changes in response to the reviewers' comments. In addition, I included a marked-up copy of the manuscript highlighting changes made to the original version.

Overall, the reviewers' comments were constructive, and I appreciate the feedback on the original manuscript. After addressing the issue raised, the quality of the manuscript has improved.

Below are the actions I have taken to address the concerns raised.

Best,

Wardell Powell

Editor Comments:

1. Please ensure that your manuscript meets PLOS ONE's style requirements. 

I have edited the manuscript to reflect PLOS ONE’s style requirements.

2. Please change "female” or "male" to "woman” or "man" as appropriate.

Please see pages 4 and 10 of the manuscript.

3. Please provide additional details regarding participant consent.

Please see page 8 under the method section of the manuscript.

a. Please clarify the sources of funding (financial or material support) for your study. 

The author received no specific funding for this work. 

b. State what role the funders took in the study. 

I did not work with any funders. Therefore, no funders had no role in study design, data collection and analysis, decision to publish, or preparation of the manuscript.

c. If any author received a salary from any of your funders, please state which authors and which funders.

The author received no specific funding for this work.

5. In your Data Availability statement, you have not specified where the minimal data set underlying the results described in your manuscript can be found.

Please see Appendix for minimal data set.

6. Please amend your manuscript to include your abstract after the title page.

Please see page 2 of the manuscript.

Reviewers' comments:

Reviewer's Responses to Questions

Comments to the Author

1. Is the manuscript technically sound, and do the data support the conclusions?

Reviewer #1: Partly

Reviewer #2: Yes

I added to the literature review, changed the research questions, and provided more clarity to the data analysis section of the manuscript. These edits should be enough to shift reviewer # 1 views from partly to yes.

2. Has the statistical analysis been performed appropriately and rigorously?

Reviewer #1: N/A

Reviewer #2: Yes

As reviewer # 1 indicated, statistical analysis doesn’t apply to this study.

3. Have the authors made all data underlying the findings in their manuscript fully available?

Reviewer #1: Yes

Reviewer #2: Yes

4. Is the manuscript presented in an intelligible fashion and written in standard English?

Reviewer #1: Yes

Reviewer #2: Yes

5. Review Comments to the Author

Reviewer #1: In my opinion the research questions are out of alignment with the main story of the paper. The research questions seem to be what the students are studying however I think the real story is what the students were thinking about, and their proficiency in making reasoned claims with evidence using the data provided. Also, did this exercise help the students understand the need for using data to inform decision-making? There is a good amount of literature on a justice orientation in science education that can be included as part of the theoretical framework. The author provides a good background and placing the pandemic in context, as well as identifying the need for a study by highlighting the skepticism and questionable claims made by public figures. I would encourage the author to reconsider the main message from the study. In my mind the study should focus on the students’ perspectives and their thoughts rather than determining the rela3onship between the indicators present.

I revised the research questions and added justice orientation in science to the theoretical framework. See the revised research questions below:

Research Question 1. 

What do middle school students think about when asked to explain the relationships between health in Massachusetts communities and the COVID-19 infection rates?

Research Question 2. 

How proficient are middle school students in using data on COVID-19 infection rates to make evidence-based claims of its relationship to the wealth-health gap among communities in Massachusetts?

Research Question 3. 

Did the students' investigations of COVID-19 infection rates in Massachusetts cities and towns impact their understanding of the need for using data to inform decision-making?

Additionally, I have added to the theoretical framework (see pages 5-6 of the manuscript). 

Reviewer #2: - Sampling technique is required along with the actual size of sample.

- this paper is more of qualitative research realm; so in results, discussion the data collected from focused group interview needs to be given more emphasis than document analysis. - the real-life situations and challenges of the sample need to be highlighted more; this would be a better ground to

give suggestive measures in conclusion part. - the paper is nicely drafted.

The research was indeed qualitative. Data were collected primarily through students' written feedback to assigned questions and class discussions on the students’ findings. Though interview data were not used, I expanded the theoretical framework, methods, data generation, data analysis, and the result and discussion section of the manuscript. I believe these edits satisfied the feedback received from reviewer # 2.

---

## [Editor Report · Decision Letter 1]

3 Jan 2023

A socioscientific issues approach to ninth-graders’ understanding of COVID-19 on health, wealth, and educational attainments

PONE-D-22-19529R1

Dear Dr. Powell,

We’re pleased to inform you that your manuscript has been judged scientifically suitable for publication and will be formally accepted for publication once it meets all outstanding technical requirements.

Kind regards,

Ana Larissa Gomes Machado, Ph.D

Academic Editor

PLOS ONE

---

## [Editor Report · Acceptance letter]

3 Feb 2023

PONE-D-22-19529R1 

A socioscientific issues approach to ninth-graders’ understanding of COVID-19 on health, wealth, and educational attainments 

Dear Dr. Powell:

I'm pleased to inform you that your manuscript has been deemed suitable for publication in PLOS ONE. Congratulations! Your manuscript is now with our production department. 

Kind regards, 

on behalf of

Dr. Ana Larissa Gomes Machado 

Academic Editor

PLOS ONE